# Variability among the Isolates of Broad Bean Mottle Virus and Encapsidation of Host RNAs

**DOI:** 10.3390/pathogens11070817

**Published:** 2022-07-21

**Authors:** Nipin Shrestha, Melvin R. Duvall, Jozef J. Bujarski

**Affiliations:** Department of Biological Sciences and Plant Molecular and Bioinformatics Center, Northern Illinois University, DeKalb, IL 60115, USA; duvallmel@gmail.com

**Keywords:** plant RNA viruses, bromoviruses, RNA genome variability, cellular RNA encapsidation, phylogeny

## Abstract

Broad bean mottle bromovirus infects legume plants and is transmissible by insects. Several broad bean mottle virus (BBMV) isolates have been identified, including one in England (isolate Ba) and five in the Mediterranean countries: Libya (LyV), Morocco (MV), Syria (SV), Sudan (TU) and Tunisia (TV). Previously, we analyzed the nucleotide sequence of the Ba RNA and here we report on and compare it with another five Mediterranean variants. The RNA segments in the latter ones were extensively homologous, with some SNPs, single nucleotide deletions and insertions, while the number of mutations was higher in isolate Ba. Both the 5′ and 3′ untranslated terminal regions (UTRs) among the corresponding RNAs are highly conserved, reflecting their functionality in virus replication. The AUG initiation codons are within suboptimal contexts, possibly to adjust/regulate translation. The proteins 1a, 2a, 3a and coat protein (CP) are almost identical among the five isolates, but in Ba they have more amino acid (aa) substitutions. Phylogenetic analysis revealed the isolates from Morocco and Syria clustering with the isolate from England, while the variants from Libya, Tunisia and Sudan created a different clade. The BBMV isolates encapsidate a high content of host (ribosomal and messenger) RNAs. Our studies present BBMV as a useful model for bromoviruses infecting legumes.

## 1. Introduction

The broad bean mottle virus is a member of the genus *Bromovirus* and infects broad bean (*Vicia faba*) and other eudicotyledonous plants, such as *Chenopodium* spp., *Nicotiana benthamiana* or *Pisum sativum* [1,2]. BBMV infection shows systemic mottling symptoms and is transmissible by insects [2,3]. There are several BBMV strains (isolates) known, one reported in England [4] while the others are in Europe, Africa and Asia. Specifically, several isolates of varying severity were found in Mediterranean countries. However, their serological properties are indistinguishable [2,3]. 

In our laboratory, the genome of the isolate Ba was previously sequenced, displaying three genomic RNAs and a subgenomic RNA (sgRNA4) [5,6]. Interestingly, unlike in the other *Bromoviridae*, the 5′ terminal nucleotide in all of the Ba RNAs is adenosine, with its biological significance not being known [5,6]. By using the full-length cDNA clones, the RNA2-derived defective interfering (DI) RNAs were characterized in Ba; that interfered with the virus accumulation and the symptoms of BBMV infection [7]. Despite the initial molecular characterization of the BBMV, the *Brome mosaic bromovirus* (BMV) dominated plant molecular virology as a cosmopolitan positive-sense model RNA virus. However, biological features, such as high multiplicity in host cells, ease of propagation and the host range (including *Arabidopsis thaliana*), can make BBMV a more convenient model, especially for viruses infecting legumes [3]. 

The purpose of the study is to sequence the full genome of five BBMV isolates collected from Libya, Morocco, Syria, Sudan and Tunisia, and characterize the genetic variability among these five isolates, based on their full genome sequences, comparing them with the previously sequenced isolate from England. 

## 2. Results and Discussion

### 2.1. Propagation and Purification of BBMV RNAs

The BBMV isolates were grown in *Vicia faba* seedlings, and all induced the systemic mottling symptoms, with the isolate TU being the most severe, whereas the isolates LyV, MV and TV caused mild mottling, and SV was the least symptomatic. The visual inspection of the symptoms indicated that for some of the isolates, these observations differed from previous studies, which showed that isolates MV and TV were among the most severe and that isolate TU was the least severe [2,3]. The virions were extracted by cesium chloride (CsCl) gradient centrifugation, and the potentially occluding cellular nucleic acids were removed by nucleases (see Materials and Methods). The presence of the purified virions was confirmed by transmission electron microscopy, as shown in Appendix A (Appendix A). Subsequently, the viral RNAs were extracted from the purified virions, and denaturing gel electrophoresis showed the presence of the intact BBMV RNA segments, RNAs1+2, RNA3 and sgRNA4 (Figure 1). 

### 2.2. Sequencing with the NGS and Sanger Protocols

The extracted BBMV RNAs were sequenced as described in Section 4. The Sanger sequencing of the 5′ RACE product corroborated the 5′ terminal A residues in all of the BBMV RNAs, previously reported for isolate Ba [5,6]. The other members of the bromoviruses are known to carry the 5′ terminal G nucleotide [5,8,9]. Other examples of the 5′ A (or U) among plant RNA viruses involved members in several families [5]. The difference in the 5′ nucleotide, the substrate for the guanidyl transferase, might hint towards specific requirements for the capping mechanism in BBMV [5,6]. The sequences of RNAs 1, 2 and 3 from the five BBMV isolates were deposited in the Gene Bank, under accession numbers, OM287444 to OM287458.

#### 2.2.1. The Particle Heterogeneity

The particle heterogeneity was analyzed, based on the number of reads (Appendix A, Appendix A) that were mapped to RNA1, RNA2 and RNA3, assuming that in bromoviruses the respective RNA components are packaged into separate virions [10]. The RNA1-bearing particles were the most abundant in the four isolates, LyV, MV, SV and TU, followed by RNA2 and RNA3, whereas in isolate TV, the RNA2 segments were the most abundant, followed by RNA1 and RNA3. In isolate SV, the difference in the RNA1 to RNA2 ratio was the highest among the five isolates (Figure 2; Appendix A). Such a heterogeneous distribution of particles in the infected plants might influence the disease progression, including the differences in symptom severity among the different isolates shown by Makkouk et al. (1989) [3]. In brome mosaic virus (BMV), the host–RNA interactions led to particle heterogeneity that affected the timing of infection, due to an altered dissociation of CP to release the viral RNAs [10,11].

#### 2.2.2. Polymorphism at the Nucleotide Level

The sequence data confirmed the general organization of the BBMV RNA genome in all of the isolates, with both RNAs 1 and 2 being monocistronic whereas RNA3 encoded two proteins. The five Mediterranean isolates were similar, at 98% for RNA1 and RNA2, while RNA3 was at 93–95% (Figure 3). However, after aligning to the isolate Ba, the RNAs appeared to be different, with increased SNPs. In particular, the alignments revealed single-nucleotide polymorphisms (SNPs) and micro-indels (1–3 nts), with RNA1 carrying 72 SNPs and a single nucleotide deletion, RNA2 carrying 64 SNPs and a single nucleotide deletion, while RNA3 was carrying 90 SNPs and 28 indels. The total SNP numbers increased to 182, 376 and 230 for RNA1, RNA2 and RNA3, respectively, after comparing with the isolate Ba.

As in Ba, the AUG initiation codons in all of the RNAs were flanked by sequences differing from the optimal consensus motifs, CCACCAUGG or AACAUGG. Similar trends were observed for the AUG codons in BMV and Cowpea chlorotic mottle bromovirus (CCMV) RNAs [5,6]. The suboptimal AUG contexts might self-limit/regulate the viral protein synthesis, and/or allow for parallel translation from downstream AUG codons, as observed in the BMV CP open-reading frame (ORF) [12].

#### 2.2.3. The Noncoding Regions

The intergenic regions, 5′ and 3′ UTR, were conserved on a particular RNA segment among the different isolates. The isolates (including Ba) shared 65 nts of 100% identity in 5′ UTR of RNA1; the 5′ UTRs in RNA1 were similar in length among BMV, BBMV and CCMV. In RNA2, the 5′ UTR was 109 nucleotides (nts) long, 100% identical in the four isolates, but in SV there was a mutation at the 64th position; in isolate Ba, this region was 110 nts long and had five SNPs. The first 30 bases in RNA2 were similar among the three bromoviruses, BMV, BBMV and CCMV [6], likely as part of the RNA replication signal [13,14].

The 3′ UTR in RNA1 was 193 nts in all five of the BBMV isolates and was not identical among any of the isolates. The least identity was between the LyV and MV isolates (97.41%), with four SNPs and one indel. This non-coding region was 272 nts for BMV and 223 nts for CCMV. These differences are due to deletions of the largely pseudo-knotted [15,16] sequences (62 and 53 nts, respectively) immediately after the 1a termination codon of BBMV and CCMV, revealing some tolerance to the mutations in the promoters of the minus-strand synthesis [5]. The RNA2 had the longest 3′ UTR of 320 nts. The isolate SV had the most variation in this region, with six SNPs and a deletion.

The three non-coding regions of BBMV RNA3 included the 5′ UTR, 3′ UTR and the intergenic region (IGR). In contrast to RNA2, the 5′ UTR of RNA3 contained 328 nts, compared to 91 nts in BMV and 238 nts in CCMV. Their first 116 nts were identical in the five BBMV isolates, but the rest carried seven SNPs. Regarding the 3′ UTR, which was 238 nts long, the SV and MV isolates shared 100% identity, but the other isolates had 13 SNPs. The most polymorphic region was the IGR, which was attributed to the varied length of the homopolymeric poly A tract, and averaged between 68 nts for isolate Ba [6], and 61, 80, 51, 54 and 55 nts for the isolates LyV, MV, SV, TU and TV RNA3s, respectively. In BMV and CCMV, the poly A tracts averaged 20 and 40 nts, respectively. The molecular function(s) of the poly A tracts have not been verified in these viruses [8,9].

Besides the poly A variation, the rest of the IGR was largely conserved in BBMV, with only 10 SNPs among the five isolates. In BMV and CCMV, the sg promoter consisted of a “core” sequence that preceded the sgRNA4 initiation site. The corresponding initiation site in the five new isolates aligned with the previously mapped site in Ba [6], but with some variation within the core (Figure 4), that might affect the transcription of sgRNA4.

#### 2.2.4. Polymorphism at the Amino Acid Level

The open reading frames in respective RNAs were translated and aligned as shown in Figure 5 and Appendix A. The SNPs in the ORF were mostly synonymous, having little effect on the amino acid composition in the five new isolates, but in isolate Ba there were more mutations. The conserved domains were found in both the 1a and 2a proteins (Figure 5, Appendix A). Two domains in 1a included the N- (amino acids 1 to 432) and C-terminal (amino acids 572 to 959) regions, comprising methyl transferase and helicase domains, respectively, as in the other bromoviruses [15]. For the 2a protein, the sequences of the viral RNA-dependent RNA polymerase (RdRp) were 100% identical between the isolates MV and TU, whereas in the isolates LyV, SV and TV there were one, two and five amino acid substitutions, respectively. The central domain carried the conserved Glycine–Aspartic acid–Aspartic acid (GDD) motif (Appendix A), a catalytic site of the RdRp enzyme [16,17], implying similar interactions, revealed in the crystal structures of BMV and CCMV [18,19].

Among the five isolates, the least homology was found for 3a and CP (Appendix A). The polymorphism in 3a was related to symptom severity by different isolates [20,21,22]. A 100% identity was found among the N-terminal LAGLI-like domains (positions 147–151) [6], that maintain the proper conformation of 3a [23]. The similarity among the 3a proteins varied between 65% with BMV vs. CCMV and 59% with BMV vs. BBMV; possibly a factor for their host range. The similarities for the 1a and for 2a proteins between BMV and CCMV also varied [5,6,9], which together might specify varied host ranges in the three bromoviruses. Along these lines, the swapping of 1a and 2a ORFs between BMV and CCMV confirmed their best fit for multiplication in these viruses’ cognate hosts [15,24].

The CP was highly similar among the five BBMV isolates, with 100% similarity between the isolates LyV and TU. The isolate TV had a substitution 46-T to A, and MV and SV shared 48-T to the S substitution, whereas isolate Ba carried 12 aa substitutions (Appendix A). The well-studied arginine-rich N-terminal domain was conserved, except in the isolate Ba, where a 9-T was replaced with A and 21–25 RQLAL was replaced with SNRLR. The RQLAL motif plays a role in the selective packaging of sgRNA4 and during RNA replication in BMV and CCMV [16,17]. The observed changes within CP might affect the electrostatic balance in the virions and the infectivity [3], as shown for BMV [25,26]. In numerous viruses, the N-ARM extends inwards and directly interacts with the RNA motifs [26], affecting the infection process. The overall homologies among the bromovirus CP sequences covered mostly the hydrophobic amino acids.

### 2.3. Phylogeny of BBMV Variants

The presence of the subgenomic promoter-like sequences only within 5′ UTRs of BBMV and CCMV RNA3 suggested that BMV RNA3 was derived by deletion from the predecessors carrying an undivided genome [6]. Among the three bromoviruses, the higher similarity between the BMV and CCMV proteins, as well as the unique 5′ A residue in BBMV, indicated the existence of an intermediate ancestor [5,6,9]. In fact, both BBMV and CCMV can infect eudicotyledonous plants, whereas BMV prefers monocots. To analyze their evolutionary relationships, the consensus cladogram was generated (Figure 6). The RNA sequences were concatenated as one sequence per isolate and aligned together, from which a cladogram was inferred by the maximum likelihood bootstrap method in Geneious, with 100 bootstrap replicates. The concatenated BBMV lineages gave bootstrap values >90. The resulting tree (Figure 6) revealed two sister clades, each with a different ancestor, but not correlating to the geographical distribution, e.g., the Morocco and Syria isolates were geographically separated, but were more closely related.

### 2.4. BBMV Encapsidate Host RNAs

Previously, we demonstrated that BMV can encapsidate nearly 0.1% of host (cellular) RNAs, including rRNAs and mRNAs, nuclear and organellar, transposable elements and retro-transposons [27]. Here we find that the BBMV packages a 10 to 20 times (between 1.14 to 1.95%) higher content of the host RNAs. Appendix A (Appendix A) summarizes the numbers of reads mapped to various categories, whereas Appendix A (Appendix A) shows that over 99% of the host reads were of nuclear origin; in BMV, 50% of the reads were from nuclear RNAs and the rest from mitochondria and plastids (Figure 7; Appendix A). Among the most mapped total 17 transcripts (Appendix A, Appendix A), although there was some variability in their concentration (based on the number of reads), five isolates encapsidated the photosystem repair protein and ribosomal protein S25, followed by aldolase, heat shock and GTP-binding proteins; the transposable elements (TEs) included retrotransposons Class 1 (LINE and LTR), and some of the Class 2 Res (Appendix A). One can speculate that these differences originate either due to the distinct affinities of viral CPs with the RNA types, due to different replication sites, or because of host-specific factors. It seems that the encapsidation in the different bromoviruses might use different specificity-based mechanisms of RNA packaging. The study of the host RNA encapsidation by BBMV will be extended further to address the subject in more detail. Future analyses, including the isolate Ba, will involve the validation of host RNAs by Northern blotting, and the possibility of artifacts from residual RNAs after nuclease treatment.

## 3. Summary and Conclusions

Our data added new information about the genus *Bromovirus*, with five new sets of BBMV RNA genomes sequenced (OM287444 to OM287458). Assuming the genome size of BBMV as nearly 8600 nts, each nucleotide was covered at the depth of 7925 reads, or 1.2 × 10^6^ nts of reads/per reference nt. This, however, may vary, as the NGS coverage depends on multiple factors [28]. In this work, the accuracy of the NGS data was confirmed by Sanger sequencing.

These results provide the complete genome and characterization of the BBMV variants, which can be used to study the biology of bromoviruses. All of the BBMV RNAs carry the 5′ terminal A, which might reflect some distinct function(s) of the 5′ cap structure in BBMV; an interesting subject for future studies. Secondly, a high similarity among both the nts and protein sequences of BBMV suggests that the previously described variations in symptom severity among the isolates [3] might be due to changes in the selected domains, working as symptom determinants, to be mapped. Third, in the same way as BMV, the BBMV isolates encapsidate the host RNAs, but their range differs, which may reflect the distinctive packaging mechanism(s). Finally, the sequences among the known BMV strains differ to a higher degree than those among the five BBMV variants. Based on the criteria defining viral strains [29], we defined the Mediterranean variants as isolates, whereas Ba BBMV should be considered as a separate strain. BMV and CCMV have been studied for host–virus interactions [16,30,31,32,33], and used in nano-biological applications [34,35]. Our findings on the genome of BBMV for different variants will contribute towards the studies of the different molecular aspects of the virus, and establish BBMV as another model Bromovirus with which to study plus-sense RNA viruses

## 4. Materials and Methods

### 4.1. Virus Material and Purification

The Mediterranean isolates of BBMV were provided by Dr. Safaa Kumari (International Center for Agricultural Research in the Dry Areas (ICARDA), Beirut, Lebanon). The viruses were propagated in the broad bean leaves, followed by extraction and purification, as described before [27,36]. Briefly, the cell extract was ultracentrifuged on a sucrose cushion and cleaned by nuclease treatment to remove any occluding RNAs/DNAs [27,37]. The nucleases were removed on Amicon Ultra-4 centrifugal filters, Ultracel-100K (UFC810008), and the washed virus suspension was further purified in 40% (w/v) CsCl gradient. The virion bands were collected and dialyzed against the virus buffer. The quality of the purified BBMV virions (Appendix A, Appendix A) was examined in a transmission electron microscope, as described previously for BMV [27].

### 4.2. RNA Extraction and NGS Sequencing

The virions were lysed with 0.5% (w/v) sodium dodecyl sulfate (SDS), and the RNA purified by phenol-chloroform extraction in a buffer 0.5 M glycine, 0.5 M NaCl, 0.1 M EDTA, pH 9.5., followed by ethanol precipitation. The RNA integrity was analyzed in 2100 Agilent Bioanalyzer, Santa Clara, CA, USA (Figure 1). The NGS RNA-Seq sequencing was completed at the University of Illinois at Chicago, Research Resource Center—DNA Services Facility. The sequencing libraries were prepared by the Illumina Truseq stranded library preparation kit. The libraries were sequenced on an Illumina NextSeq 500 instrument, paired end 2 × 150 (Illumina, San Diego, CA, USA).

### 4.3. Rapid Amplification of cDNA Ends (RACE) and Sanger Sequencing

The 3′ and 5′ termini were determined by PCR amplification of the poly-A enriched RNAs into full-length cDNAs with the SMARTER 5′/3′ RACE kit, Takara Bio USA Inc., CA, USA, using the designed RNA-specific primers. The sequences were confirmed by Sanger sequencing at Eurofins genomics, Louisville, KY, USA. The remaining regions were also sequenced with the Sanger protocol, with pairs of primers overlapping over at least 30 nts [38]. This confirmed the NGS assembly by over 99%. The exact length of the poly A stretches in RNA3 were not determined, due to the limitations of the Sanger protocol [38,39].

### 4.4. De Novo Genome Assembly and Sequence Comparisons

The NGS reads were quality trimmed to the minimum length of 25 bp, using the DynamicTrim and LengthSort programs from SolexaQA [27,40], and were assembled into a representative sequence of the population [41,42] using the Iterative virus assembler (IVA) pipeline [43]. The IVA has an option of the Trimmomatic program [44] that trims off the adapters and primers before the assembly process. The contigs were quality-checked by the IVA QC in a default setting, then queried against the nucleotide database, using BLASTn (https://blast.ncbi.nlm.nih.gov/Blast.cgi, accessed on 24 May 2022). The verified sequences were subjected to further comparisons, using the Geneious 10.2.3 (Biomatters, Ltd., Auckland, NZ, USA). The contigs that mapped to the previously sequenced BBMV RNAs from the Ba isolate were extracted to the length of the reference by using bedtools2 [45], and aligned to the GenBank references for BBMV RNA1 (NC_004008), RNA2 (NC_004007), and RNA3 (NC_004006) using Geneious 10.2.3.

The open reading frames for the respective RNAs were identified by the ORF finder in Geneious and translated by using standard codons. The sequence similarities and conserved motifs in the translated proteins were aligned with the MAFFT and MUSCLE sequence alignment tools.

### 4.5. Phylogeny Analyses

The BBMV RNA1, RNA2 and RNA3 were concatenated to generate a single RNA molecule of 8394 nts per each of the five Mediterranean isolates, whereas 8250 nts were generated for the England isolate. BMV was used as an outgroup to infer the rooted cladogram. The RNA1 (NC_002026.1), RNA2 (NC_002026.1) and RNA3 (NC_002026.1) from BMV were also concatenated into a single RNA molecule of 8201 nts. The maximum likelihood bootstrap method in Geneious, with 100 bootstrap replicates, was used to reconstruct the consensus tree.

## Figures and Tables

**Figure 1 pathogens-11-00817-f001:**
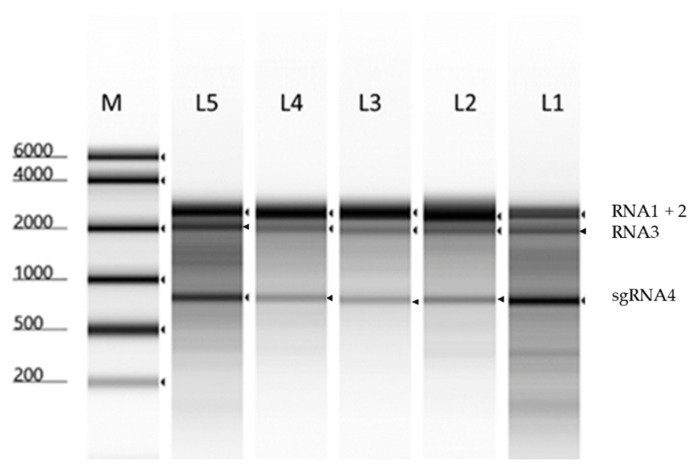
Denaturing gel of the BBMV RNAs isolated from five BBMV isolates. The images were taken prior to sequencing. L1, L2, L3, L4 and L5, which are the lanes loaded with the RNA from MV, LyV, SV, TU and TV isolates, respectively. Lane M is the size marker. The three major bands, characteristic for Bromoviruses, representing RNA1 + 2 (~2900–3100 bp), RNA3 (~2200 bp) and sgRNA4 (~800 bp) are visible in the gel (indicated on the right side).

**Figure 2 pathogens-11-00817-f002:**
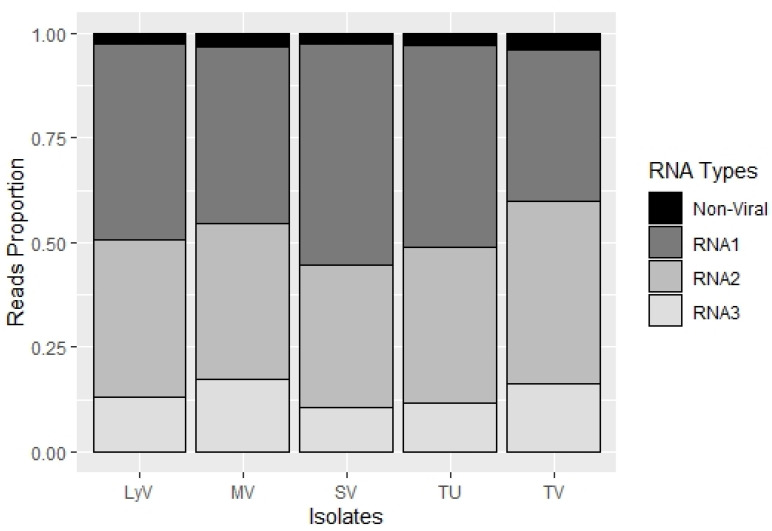
A stacked plot showing proportion of reads that mapped to RNA1, RNA2 and RNA3 among the five BBMV isolates.

**Figure 3 pathogens-11-00817-f003:**
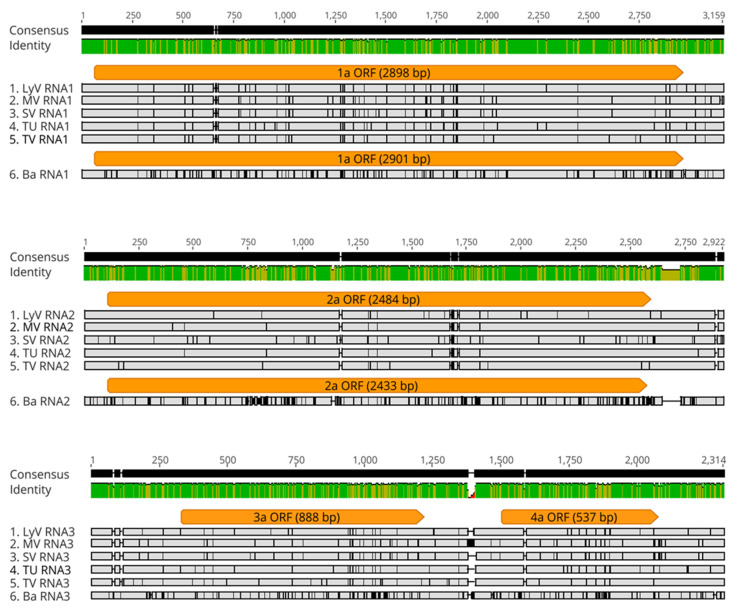
Multiple sequence alignment (MSA) of nucleotide sequences of BBMV RNAs 1, 2, and 3 (top to bottom) for five BBMV isolates as compared to Ba ones. The orange bars represent the ORF regions (with the length shown in parentheses). The gray regions signify identical nucleotides, while the nucleotide substitutions are depicted by the black vertical lines and the deletions by the horizontal black lines. The green bars illustrate the regions of identity.

**Figure 4 pathogens-11-00817-f004:**
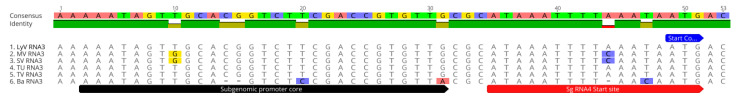
The alignment of the nt sequences of the intercistronic noncoding region in the RNA3 of the BBMV variants covering the subgenomic promoter core (black bar) and the start site of the sg RNA4 transcription (see also Romero et al., 1992 [6]). The subgenomic RNA start site are marked by red bar, and the CP translation start codon ATG is indicated by the blue bar on top of the alignment.

**Figure 5 pathogens-11-00817-f005:**
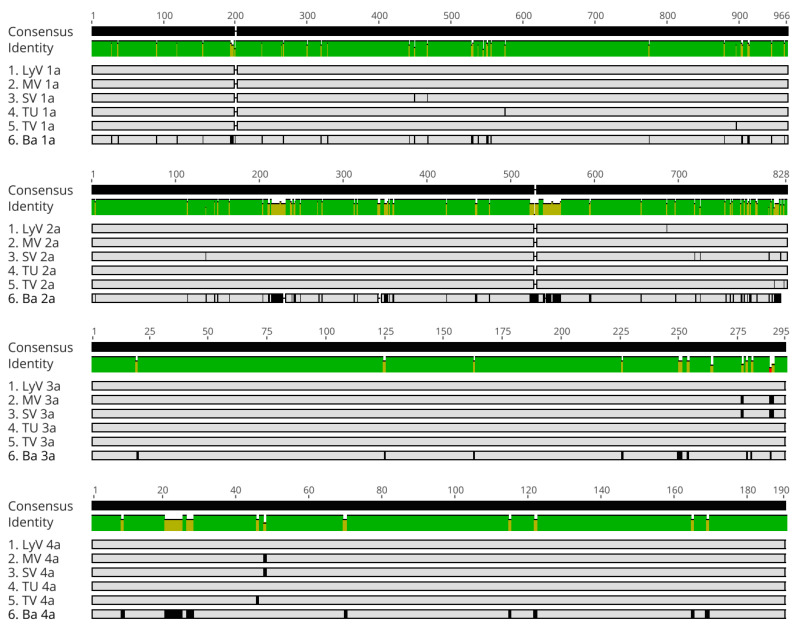
MSA of amino acid sequences of proteins 1a, 2a, 3a and 4a (CP) for six BBMV isolates. The gray regions represent identical nucleotides, while the aa substitutions are depicted by the black vertical lines and the deletions by small gaps. The green bars illustrate the regions of identity. The numerical scale on top marks the positions of amino acids.

**Figure 6 pathogens-11-00817-f006:**
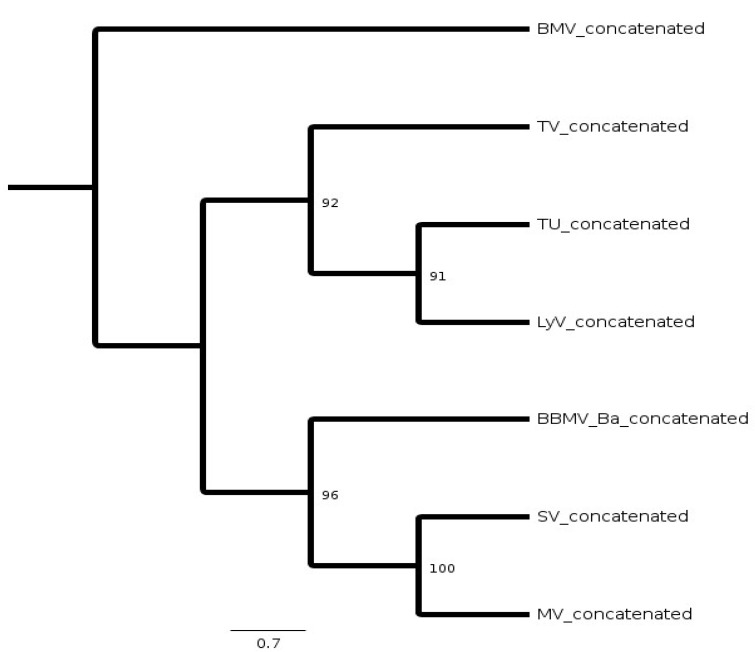
Maximum likelihood (ML) bootstrap consensus cladogram for five Mediterranean and the England isolates of BBMV. The sequences of RNAs 1, 2 and 3 were concatenated and aligned by MSA followed by the consensus tree reconstruction by the ML bootstrap method. The BMV RNA sequences were used as the outgroup to infer the rooted cladogram. Numbers along the branches are ML bootstrap values. The cladogram has been calculated at the Geneious default values.

**Figure 7 pathogens-11-00817-f007:**
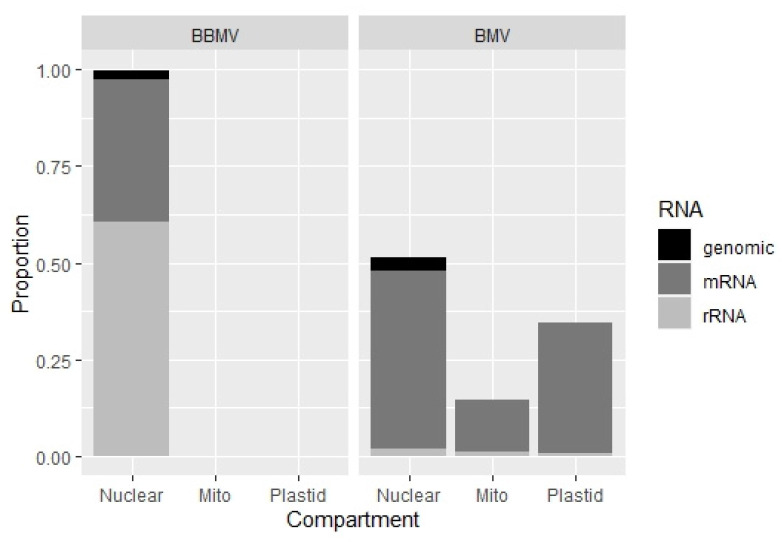
The stacked plot illustration profiling the proportions of different types of encapsidated host RNAs in BBMV vs. BMV, originating from three different cellular compartments. Apparently in BBMV, more than 99% of the host RNAs were nuclear RNAs with ribosomal RNAs being present more in comparison to the mRNA transcripts, whereas in BMV over 50% of the reads mapped to the nuclear RNAs followed by the organellar RNAs (mitochondrial and plastid). The proportion of the mRNAs being mapped dominantly in comparison to the rRNAs also differed between BBMV and BMV.

## Data Availability

The genome assembled for the five isolates of BBMV in this project are deposited in the NCBI GenBank under accession: OM287444 to OM287458.

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
