# Peer review of "Variability among the Isolates of Broad Bean Mottle Virus and Encapsidation of Host RNAs"

_pathogens, 2022, doi:10.3390/pathogens11070817_

Round 1
Reviewer 1 Report
This manuscript is a resubmission of a previously submitted report. The work consists of the sequencing and basic genetic analyses of the sequences of five African Broad bean mottle virus (BBMV) isolates. The work is complemented by analyses of NGS data of purified virus particles. TEM of virus particles of BBMV isolates demonstrated the purification. The most interesting and sound results is the finding of encapsulation of nuclear plant RNAs (genomic mRNAs, rRNAs) with the BBMV virus particles, in contrast with brome mosaic virus (BMV) that encapsidates also mitochondrial and chloroplast RNAs. Finally, the manuscript is well written.
Author Response
This manuscript is a resubmission of a previously submitted report. The work consists of the sequencing and basic genetic analyses of the sequences of five African Broad bean mottle virus (BBMV) isolates. The work is complemented by analyses of NGS data of purified virus particles. TEM of virus particles of BBMV isolates demonstrated the purification. The most interesting and sound results is the finding of encapsulation of nuclear plant RNAs (genomic mRNAs, rRNAs) with the BBMV virus particles, in contrast with brome mosaic virus (BMV) that encapsidates also mitochondrial and chloroplast RNAs. Finally, the manuscript is well written.
Thank you for reviewing the paper. We are glad you found our work important and publishable.
Reviewer 2 Report
During the previous review it had been asked for:
1. Though authors demonstrated encapsidation of host RNAs, it is essential to validate by another method such as Northern blot.
2. Further, the study shows that the important host transcripts are encapsidated (as presented in Table 1). It would be nice to see whether the encapsidation of transcripts is isolate-dependent or not? Additionally, all the isolated used for encapsidation study are from the Mediterranean. It is worth including isolate Ba.
These are not addressed in the revised version. Additionally, I am unable to find a response to the previous clarifications.
Author Response
During the previous review it had been asked for:
Though authors demonstrated encapsidation of host RNAs, it is essential to validate by another method such as Northern blot.
The validation by other methods as e.g. by Northern blotting will be done in the future and such sentence has been added to the text describing the future research plans on this subject.
Further, the study shows that the important host transcripts are encapsidated (as presented in Table 1). It would be nice to see whether the encapsidation of transcripts is isolate-dependent or not? Additionally, all the isolated used for encapsidation study are from the Mediterranean. It is worth including isolate Ba.
>>The question about if the encapsidation of transcripts is isolate-dependent has been, in part, already addressed in Table 4S and these studies will continue as future research plans, including the more different isolate Ba, especially that Ba carries much more altered amino acids in its CP than other isolates do (see Fig. 5).
These are not addressed in the revised version. Additionally, I am unable to find a response to the previous clarifications.
Reviewer 3 Report
Comments in attached file.

Author Response
Please find the attachment

This manuscript is a resubmission of an earlier submission. The following is a list of the peer review reports and author responses from that submission.
Round 1
Reviewer 1 Report
Comments are in the attached file.

Reviewer 2 Report
It was a pleasure reviewing the article entitled “Variability among the isolates of Broad bean mottle virus and encapsidation of host RNAs” submitted for consideration in the Pathogens. In the study, complete nucleotide sequences of 6 Broad bean mottle bromovirus (BBMV) isolates were subjected to comparative research to understand the genetic variability. Though authors demonstrated encapsidation of host RNAs, it is essential to validate by another method such as Northern blot. Further, the study shows that the important host transcripts are encapsidated (as presented in Table 1). It would be nice to see whether the encapsidation of transcripts is isolate-dependent or not? Additionally, all the isolated used for encapsidation study are from the Mediterranean. It is worth including isolate Ba.
Other comments are in the attached .pdf

Reviewer 3 Report
This manuscript describes the genetic characterization of five broad bean mottle bromovirus (BBMV) isolates from five different countries. Virions were isolated for RNA extractions and NGS sequencing. Draft genomic sequences were obtained from NGS as was followed by 5’RACE and the sequencing of the 3’end after polyadenilation of the RNA followed by PCR with a polyT and specific primers. Sanger sequencing allowed the confirmation of the genomic sequences. Phylogenetic and genetic analysis were then carried out.
In my opinion, although the manuscript is well written, there are no significant novelties in both the results and technical approaches that, in my opinion, makes the manuscript sufficient for a high impact journal. These results may be of interest to a local or regional journal.